# Coupling Square Wave Anodic Stripping Voltammetry with Support Vector Regression to Detect the Concentration of Lead in Soil under the Interference of Copper Accurately

**DOI:** 10.3390/s20236792

**Published:** 2020-11-27

**Authors:** Ning Liu, Guo Zhao, Gang Liu

**Affiliations:** 1Key Laboratory of Modern Precision Agriculture System Integration Research, Ministry of Education of China, China Agricultural University, Beijing 100083, China; ningliu@cau.edu.cn; 2Key Laboratory of Agricultural Information Acquisition Technology, Ministry of Agriculture and Rural Affairs of China, China Agricultural University, Beijing 100083, China; 3College of Artificial Intelligence, Nanjing Agricultural University, Nanjing 210031, China; zhaoguo@njau.edu.cn

**Keywords:** lead in soil, interference of copper, support vector regression (SVR), nonlinear relationship, bismuth film-modified electrode

## Abstract

In this study, an effective method for accurately detecting Pb(II) concentration was developed by coupling square wave anodic stripping voltammetry (SWASV) with support vector regression (SVR) based on a bismuth-film modified electrode. The interference of different Cu^2+^ contents on the SWASV signals of Pb^2+^ was investigated, and a nonlinear relationship between Pb^2+^ concentration and the peak currents of Pb^2+^ and Cu^2+^ was determined. Thus, an SVR model with two inputs (i.e., peak currents of Pb^2+^ and Cu^2+^) and one output (i.e., Pb^2+^ concentration) was trained to quantify the above nonlinear relationship. The SWASV measurement conditions and the SVR parameters were optimized. In addition, the SVR mode, multiple linear regression model, and direct calibration mode were compared to verify the detection performance by using the determination coefficient (*R*^2^) and root-mean-square error (RMSE). Results showed that the SVR model with *R*^2^ and RMSE of the test dataset of 0.9942 and 1.1204 μg/L, respectively, had better detection accuracy than other models. Lastly, real soil samples were applied to validate the practicality and accuracy of the developed method for the detection of Pb^2+^ with approximately equal detection results to the atomic absorption spectroscopy method and a satisfactory average recovery rate of 98.70%. This paper provided a new method for accurately detecting the concentration of heavy metals (HMs) under the interference of non-target HMs for environmental monitoring.

## 1. Introduction

A large amount of lead ions have accumulated in soil in recent decades due to improper agricultural management, such as excessive usage of fertilizers, pesticides and sewage water irrigation [1], and the addition of wine vinasse [2,3]. Lead, as one of the most toxic heavy metals (HMs), can cause acute and chronic damage to the eco-environment even at low concentrations and harm human health because of its spreadable accumulation through food chains [4,5]. Therefore, developing a high-efficiency and fast method for accurate detection of lead in soil is of great necessity.

Square wave anodic stripping voltammetry (SWASV), as an electrochemical analysis technology, was widely applied for the detection of HMs [6,7]. SWASV, possessing many advantages of low cost, high sensitivity, excellent selectivity, and fast detection, was a good alternative to spectroscopic analysis methods [8,9,10], such as atomic absorption spectroscopy (AAS) and inductively coupled plasma-atomic emission spectroscopy. During the analysis of HMs by using SWASV, all target HMs were electroplated onto the surface of the working electrode (WE) in the electrodeposition step and then stripped off the WE surface in the electroelution process. A variety of HMs can be recognized using stripping peak potentials, and the amplitude of stripping peak currents is proportional to the concentration of corresponding HMs [11]. However, the presence of copper ion (Cu^2+^), which was found to be the largest interference ion, would seriously hinder the stripping signals of Pb^2+^ [12], especially on the WE containing bismuth material [13], further decreasing the detection accuracy of the SWAVS method. The negative effect may be due to the formation of intermetallic Bi–Cu and Cu–Pb compounds [14,15] and the competition for active sites on the WE surface [16].

To improve the detection performance of SWASV, chemically modified electrodes (CMEs) have been investigated by numerous researchers [17,18,19]. Nevertheless, CMEs suffer from many limitations, including expensive modification materials, complex modification procedures, and rigorous storage environments, which will limit their in situ application [20]. Moreover, the presence of Cu^2+^ still obviously decreases the tripping peak current of Pb^2+^ [21]. Previous reports [13,15,21,22] indicated that the Cu^2+^ interference could be decreased to some extent by adding ferrocyanide, which could form insoluble and stable copper–ferrocyanide complexes. However, the adding amount of ferrocyanide required an optimization procedure to achieve the optimal masking effect for each sample [11], which was seriously time consuming and would decrease the detection efficiency of Pb^2+^ in soil samples.

Although Cu^2+^ would decrease the stripping peak current of Pb^2+^, SWASV could measure the stripping currents of Pb^2+^ and Cu^2+^. Accordingly, the interference law of Cu^2+^ content on the stripping peak current of Pb^2+^ could be investigated through statistical analysis by adding different concentrations of Cu^2+^. A nonlinear relationship between Pb^2+^ concentration and the peak currents of Pb^2+^ and Cu^2+^ was found in this paper. Thus, the support vector regression (SVR) was chosen to characterize the above nonlinear relationship to establish a quantitative model for accurate detection of Pb^2+^ concentration. SVR, as a learning algorithm for small sample size, is widely used to solve nonlinear regression questions [23,24]. Many scholars have indicated that the radial basis function (RBF) net is an efficient tool for the fitting of a nonlinear relationship [25] and can efficiently eliminate the overfitting problems that widely exist in traditional nonlinear regression models [26]. Therefore, in this study, the radial basis function was selected as the SVR kernel function to interpret the nonlinear relationship between the stripping peak currents of Pb^2+^ and Cu^2+^ and Pb^2+^ concentration. To avoid the limitations of complex CMEs, a bismuth film-modified glassy carbon electrode (Bi/GCE) was used to perform the SWASV measurement in this study due to the many advantages of bismuth, such as wide potential window, low toxicity, and ability to reduce the activation energy of Pb^2+^ [27,28,29].

In this study, the interference law of different Cu^2+^ contents on the stripping peak current of Pb^2+^ and the masking effect of ferrocyanide on the Cu^2+^ interference for Pb^2+^ SWASV signal measurement were investigated. Then, SVR was used to quantify and interpret the nonlinear relationship between Pb^2+^ concentration and the stripping peak currents of Pb^2+^ and Cu^2+^ and to establish a model for accurate detection of Pb^2+^ concentration. In addition, the detection performance of the SVR model was compared with that of the multiple linear regression (MLR) model and direct calibration models to confirm the above nonlinear relationship and highlight the excellent performance of the SVR model. Lastly, real soil samples were collected to verify the feasibility and practicality of the proposed method.

## 2. Materials and Methods

### 2.1. Reagents and Apparatus

All chemical reagents were of analytical grade. The working solutions of Bi^3+^, Pb^2+^, and Cu^2+^, which were prepared from 1000 mg/L of standard stock solutions of Bi(NO_3_)_3_, Pb(NO_3_)_2_, and Cu(NO_3_)_2_ (Nation Standard Materials Center of China, Beijing, China) were diluted as required. An acetate buffer solution (0.2 M), which was prepared by mixing acetic acid and sodium acetate (Beijing Noble Ryder Technology Co., Ltd., Beijing, China), was used as the electrolyte for the electrodeposition and stripping of Pb^2+^ and Cu^2+^. K_3_Fe(CN)_6_ (Shenzhen Nanotech Port Co., Ltd., Shenzhen, China) was used as a masking agent to reduce the interference of Cu^2+^. Millipore-Q water (18.2 MΩ) was used as the cleaning and dilution agents for all experiment.

In addition, a three-electrode system, consisting of a WE made of Bi/GCE, a platinum wire counter electrode, and an Ag/AgCl reference electrode, was used for the electroanalysis of lead and copper. An electrochemical workstation with CHI660D type (Shanghai CH Instruments Co., Ltd., Shanghai, China) was applied to perform the electrochemical measurements, including SWASV, cyclic voltammetry (CV), and constant potential. A magnetic stirrer with a stir bar placed into a 25 mL cell was applied to stir the solution under test during the electrodeposition and cleaning processes.

### 2.2. Preparation of Bi/GCE

A bare glassy carbon electrode (GCE) was pretreated as follows. First, the GCE was polished using 0.05 mm alumina powder on a polishing paper made of deerskin material. Second, the GCE was cleaned via ultrasound for 3 min by sequentially soaking in 50% HNO_3_, absolute ethanol, and Millipore-Q water and dried using N_2_ gas. Third, Bi^3+^ was diluted to a concentration of 60 μg/L in the 20 mL of acetate buffer solution (0.2 M, pH 5.0). Lastly, for the electrodeposition process, the pretreated GCE was placed into a beaker at a potential of −1.2 V for 150 s while stirring the solution to be tested to obtain Bi/GCE, during which the Bi^3+^ and HMs were electroplated into the GCE surface, named in situ coating bismuth film.

### 2.3. Collection of Pb^2+^ and Cu^2+^ SWASV Signals

Under the optimized conditions, SWASV measurements of Pb^2+^ and Cu^2+^ were conducted in an acetate buffer solution (0.2 M, 5.0 pH) containing 60 μg/L of Bi^3+^ on Bi/GCE. During electrodeposition, a reduction potential of −1.2 V (vs. the Ag/AgCl reference electrode) was applied to the Bi/GCE WE for a deposition time of 150 s while stirring the solution to be tested. Then, it was left for 10 s to allow the solution to reach equilibrium. Afterward, stripping voltammetry was performed in the potential range of −1.0 V to 0.2 V without stirring. During the stripping process, the frequency, step amplitude, and pulse amplitude of the square wave stripping voltage were 25 Hz, 5 mV, and 25 mV, respectively. After each SWAVS measurement, an activation treatment was carried out by applying a constant potential of 0.3 V for 120 s to remove the residual Bi, Pb, and Cu ions from the WE surface.

To study the stripping response law of Pb^2+^ in the existence of different Cu^2+^ concentrations, the concentration of Pb^2+^ was set with 10 gradients, which were 1, 5, 10, 15, 20, 25, 30, 35, 40, and 45 μg/L. The Cu^2+^ concentration was set with seven gradients, which were 0, 1, 5, 10, 15, 20, and 25 μg/L, because the peak current of Pb^2+^ was extremely small and kept decreasing when the Cu^2+^ concentration exceeded 25 μg/L. In accordance with the experiment arrangement, a total of 70 groups of sample data were used in this study to analyze the interference characteristic of copper ions and establish an SVR model for accurately detecting lead ions.

### 2.4. SVR Modeling

SVR, as a learning algorithm for smaller sample size developed from support vector machine, is widely used to solve nonlinear regression questions [23,30]. SVR, based on Vapnik–Chervonenkis statistical theory and the principle of structural risk, shows great advantages in high-dimensional pattern recognition [31]. To fit the nonlinear relationship in a training dataset, traditional regression methods usually add a high-order modified term, which will introduce serious overfitting problems. The above contradiction is properly solved using the kernel function in SVR [32]. Replacing the linear term in the linear equation with the kernel function can change the original linear term to a nonlinear one—that is, it can perform nonlinear regression. In this study, SVR was trained with the data from the SWASV spectrogram and the known concentrations of Pb^2+^ and Cu^2+^ under supervision because training the SVR model needed target property (namely, the known Pb^2+^ and Cu^2+^ concentrations).

The introduction of the kernel function achieves the purpose of increasing the dimension, and the added parameters are adjustable and optimizable to prevent overfitting questions. Many scholars have indicated that the radial basis function net is an efficient tool for the fitting of a nonlinear relationship [25,33]. Therefore, in this study, a radial basis function was selected as the SVR kernel function to discover and interpret the nonlinear relationship between the stripping peak currents of Pb^2+^ and Cu^2+^ and the Pb^2+^ concentration. For the radial basis function, two main parameters, which are the gamma (*g*) and penalty coefficient (*c*), affect the SVR performance. The value of parameter *c*, as the width of error, directly affects the generalization capability of the model [34]. The *g*, an inherent parameter of the radial basis function, determines the distribution of the data after mapping to the new eigenspace and support vector number, which affects the SVR model in training speed and prediction accuracy [35].

The grid search algorithm, as a traversal optimization algorithm, can be combined with cross validation to obtain a global optimal parameter combination [36]. Low optimization efficiency is the main drawback of the grid search algorithm because it needs to examine each parameter combination systematically, similar to traversing each grid in a large graph [37]. Hence, grid search is usually used for the parameter optimization of small samples, which is consistent with the characteristics of SVR (namely, small samples). On this basis, grid search is suitable for the optimization of *g* and *c* parameters of SVR, and the coupling of grid search and SVR may be an effective method to train the model for Pb(II) concentration detection.

SVR optimized by grid search was proposed to establish a nonlinear model for the detection of Pb^2+^ concentration in the presence of Cu^2+^. Two inputs, namely the stripping peak currents of Pb^2+^ and Cu^2+^, and one output, namely the Pb^2+^ concentration, constituted the SVR model, as shown in Figure 1.

As indicated in Section 2.3, a total of 70 samples were considered, which were divided into the training and testing datasets by using the random selection method. The training set had 50 samples, which were used to train the SVR model, and the test set had 20 samples, which were used to test the performance of the SVR model.

Given the small sample, fivefold cross validation was applied in this study. The iteration number of cross validation is commonly equal to the sample number of the training dataset. The optimal parameter combination of c and g was selected on the basis of the smallest root-mean-square error (RMSE) of the cross validation set. The multiple linear regression model was established with the same inputs as SVR to highlight the excellent performance of SVR in detecting the Pb^2+^ concentration. In the study, a SVR model, which could correct the interference of Cu(II), was proposed with the stripping peak currents of Pb(II) and Cu(II) as inputs to accurately detect the concentration of Pb(II) in the presence of Cu(II). The SVR modeling belonged to the chemometrics, which was different from direct calibration model between the stripping peak current of Pb(II) and concentration of Pb(II). Thus, the limit of detection of the SVR model could not be calculated and obtained. However, for the chemometric model (i.e., SVR model in this paper), the values of determination coefficient (*R*^2^), RMSE, and relative standard error (RSD) of the test dataset were used to evaluate the detection performance of various models.

### 2.5. Preparation of Soil Samples and AAS Measurement

Soil samples were collected from a farmland in Xiaotangshan town, Beijing, China. The soil mainly consisted of 39.9% sand, 46.6% silt, 13.5% clay [38], which was weak alkaline and contained a small amount of carbonate and humus but more mineral ions. The extract solutions of soil samples were prepared in accordance with the introduction of previous works [13,39], which could be briefly summarized as follows. First, the soil samples were dried in an oven for 2 h. Second, the dried soil samples were pulverized using a mortar and sieved through a 200 μm sieve. Third, 2 g of soil was transferred into a bottle with 30 mL of acetate buffer (0.2 M, pH 5.0), then the soil mixture was shaken in an end-over-end shaker for 16 h at room temperature. Fourth, the mixture solutions were subjected to centrifugal sedimentation for solid–liquid separation to obtain the soil extract solution and filtered with a membrane to remove microimpurities. Lastly, 20 mL of extract solution was loaded into a beaker, the pH value of which was adjusted to 5.0 by using acetic acid and sodium acetate. On the basis of the sequential extraction method proposed by Tessier [40], the soil HMs of water-soluble, exchangeable, and bound-to-carbonate fractions, which are bio-available forms and present considerable harm to humans and the environment, could be extracted through the above treatment.

The AAS analysis was performed at the Testing Center of USTB Co., Ltd. (Beijing, China). The determination of Pb(II) in the liquid samples was performed by a graphite furnace atomic absorption spectrometer (AAS 6300, Kyoto, Japan) under standard conditions [41]. The apparatus was calibrated by using standard solutions prepared by diluting a stock solution with 1g/mL Pb(II). A linear calibration curve (*R*^2^ = 0.999) between Pb(II) concentration and corresponding absorbance was obtained with Pb(II) concentrations ranging from 1 up to 35 μg/L. Then, the absorbance of soil extraction solution was measured to obtain the Pb(II) concentration. For each soil extraction sample, three replicated measurements were conducted.

## 3. Results and Analysis

### 3.1. Electrochemical Activity of the Bare GCE

To ensure the electrochemical activity of the GCE used in the subsequent work, the electrochemical property of this GCE was investigated via CV by using [Fe(CN)6]^3−/4−^ as the redox probe. The CV curve of the GCE in a mixture solution with 5.0 mM [Fe(CN)_6_]^3−/4−^ and 0.1 M KCl is shown in Appendix A. The obvious redox peaks of [Fe(CN)_6_]^3−/4−^ for the GCE demonstrated that this GCE could be used for the electrodeposition and electroelution of HMs and for the stripping signal collection of Pb(II) and Cu(II).

### 3.2. Optimization of Experimental Conditions

To obtain the optimal sensitivity for the SWASV determination of Pb(II) by using Bi/GCE, the relevant SWASV parameters, which were the pH value of the supporting electrolyte, the concentration of Bi(III) ions, deposition potential, and deposition time, were optimized, as shown in Figure 2.

Figure 2a shows the effect of pH on the stripping peak current of Pb(II), in which the peak current increased gradually in the range from 3.5 to 5.0. The reason for this phenomenon might be that, at a low pH value, substantial hydrogen ions (H^+^) tended to occupy the surface active sites of WE, and resist the electrodeposition process of Pb^2+^ ions due to electrostatic force [42]. As the pH value increased, the protons in the buffer solution decreased, resulting in the decreased competition for active sites and the increased number of free active sites, which promoted the absorption of abundant Pb^2+^ ions on the WE surface. However, the peak current gradually decreased in the range from 5.0 to 6.0. This decrease might be due to the partially soluble lead hydroxide complex formed in the buffer solution [43]. These hydroxide complexes would precipitate on the WE surface, resulting in a decrease in the number of free Pb^2+^ ions that could be attached on the WE surface. The maximum stripping peak current was obtained at pH 5.0. Consequently, the subsequent experiments were conducted at pH 5.0.

Figure 2b reveals that the stripping peak current of Pb(II) sharply increased from 0 to 20 μg/L of the Bi(III) ion concentration, gradually increased from 20 to 60 μg/L, and then decreased with the increase in the Bi(III) concentration. The reason was that Bi(III) could form a Bi-Pb alloy with Pb(II), which reduced the activation energy required for Pb(II) electrochemical reduction. Nonetheless, excessive Bi(III) would form a thick bismuth film and occupy plenty of active sites, which was unconducive to the electrodeposition of lead ions. Therefore, the optimal Bi(III) concentration was 60 μg/L. The optimal bismuth concentration selected in this research was smaller than that reported by others [14,15,16,27], which might be due to the less active sites on the bare GCE surface.

The optimization process of deposition potential ranging from −1.6 to −0.8 V was explored, as shown in Figure 2c. The largest peak current was achieved at −1.2 V for Pb(II) stripping, which was due to the fact that considerably positive potentials were insufficient to reduce Pb^2+^ ions, but considerably negative potentials would result in hydrogen evolution that influenced the electrosorption of Pb^2+^ on WE. For subsequent measurements, the potential of −1.2 V was applied in the electrodeposition process.

Given that deposition time directly determined the amount of lead ion accumulating on the WE surface, which affected the sensitivity of the SWASV detection for Pb(II), a series of deposition times from 60 to 300 s was researched, as shown in Figure 2d. The stripping peak current was almost linearly related to the deposition time. Regarding the detection efficiency in a higher sensitivity level, the time of 150 s was chosen.

In sum, the optimized experiential conditions were a pH value of 5.0, Bi(III) concentration of 60 μg/L, an electrodeposition potential of −1.2 V, and deposition time of 150 s.

### 3.3. Electrochemical Responses of Bi/GCE

A large stripping peak current was the precondition to achieve high sensitivity and detection accuracy. Figure 3 shows the SWASV signals of 10 μg/L of Pb(II) on the bare GCE and Bi/GCE in a potential range from −1.0 to 0.2 V under the optimized experimental conditions. As shown, the stripping current peak of Pb(II) on the bare GCE was small. Comparatively, a large stripping peak was obtained on Bi/GCE, which revealed that the bismuth ions could improve the HM ion stripping signal due to the formation of the bismuth–lead alloy, which could immensely reduce the activation energy required for Pb(II) electrochemical reduction and vastly enlarge the amount of Pb(II) ion accumulation on WE.

As shown in Figure 3, the peak current of Pb(II) obviously decreased after 5 μg/L of Cu(II) was added, which would reduce the sensing sensitivity and interfere with the precision for Pb(II) determination by using the SWASV technology. Therefore, exploring the interference of different Cu(II) concentrations on the stripping current peak of Pb(II), even seeking the interference regular, was of great significance to the accurate determination of Pb(II) concentration.

### 3.4. Interference of Cu(II) on the SWASV Determination of Pb(II)

Several SWASV measurements were conducted on mixtures of Cu(II) and Pb(II) to investigate the interference effect of Cu(II) in different concentrations on the stripping current of Pb(II). A group of binary mixtures of two types of HM ions, in which the Pb(II) concentration ranged from 1 to 45 μg/L and the Cu(II) concentration ranged from 1 to 25 μg/L, was prepared. Under the optimal experimental conditions, the stripping voltammograms were recorded in accordance with the SWASV measurement for the various concentration combinations of Cu(II) and Pb(II), as shown in Figure 4. As shown in embedded pictures of Figure 4a−g, the stripping current peak of Pb(II) was nearly linearly related to the concentration of Pb(II) in a specific concentration of Cu(II). However, the peak currents were obviously influenced by the Cu(II) ions even at trace content.

For Pb(II) detection by using the SWAVS method, the electrode modification with a bismuth film was usually used to enhance the sensing sensitivity, which was due to the fact that Bi(III) could alloy with Pb(II). Nevertheless, the stripping signals of Pb(II) on the electrode modified using a Bi(III) film would be severely affected by Cu(II) in accordance with previous reports [27,28,29]. In the absence of Cu^2+^, the stripping peak currents of Pb^2+^ varied from 0.47 to 30.69 μA as the Pb^2+^ concentration increased from 1 to 35 μg/L, as shown in Figure 4a. However, the stripping peak current of Pb^2+^, which varied from 0.08 to 10.59 μA, obviously decreased in the presence of 1 μg/L Cu^2+^. Subsequently, the stripping peak currents of Pb^2+^ were disturbed to varying degrees with the increase in concentration of Cu^2+^. This phenomenon could be partly attributed to the fact that Cu^2+^ possesses stronger oxidation capability than Pb^2+^ (Cu^2+^ ions are easier to reduce than Pb^2+^ ions); consequently, Cu^2+^ would preferentially form an alloy with Bi^3+^ and be electroplated on the WE surface. The above result might be caused by the formation of mixed intermetallic compounds among Pb, Bi, and Cu, which would affect the stripping peak current of one another.

The interferences of various Cu(II) concentrations ranged from 0 to 25 μg/L on the stripping current peaks of Pb(II) in the concentration range from 1 to 45 μg/L were analyzed, and the standard errors were calculated by measurements for 3 times, as presented in the form of error bars in Figure 5. The standard error of peak currents ranged between 0.11 and 0.52. The peak current of Pb(II) in the presence of 1 μg/L of Cu(II) decreased sharply compared with that in the absence of Cu(II), then it increased gradually with increasing Cu(II) concentration to 10 μg/L. The peak current of Pb(II) decreased again as the concentration of Cu(II) exceeded 10 μg/L. When the concentration of Cu(II) was more than 20 μg/L, the existence of Cu(II) still greatly reduced the peak current of Pb(II), but it tended to be relatively stable. Specifically, in the absence of Cu^2+^, the stripping peak currents of Pb^2+^ varied from 0.47 to 30.69 μA as the Pb^2+^ concentration increased from 1 to 35 μg/L, as shown in Figure 4a. However, the stripping peak current of Pb^2+^, which varied from 0.08 to 10.59 μA, obviously decreased in the presence of 1 μg/L Cu^2+^, as shown in Figure 4b. Subsequently, the stripping peak currents of Pb^2+^ was disturbed to varying degrees with the increase in concentration of Cu^2+^. This phenomenon could be partly attributed to the fact that Cu^2+^ possesses stronger oxidation capability than Pb^2+^ (Cu^2+^ ions are easier to reduce than Pb^2+^ ions); consequently, Cu^2+^ would preferentially form an alloy with Bi^3+^ and be electroplated on the WE surface. The above result might give rise to the formation of mixed intermetallic compounds among Pb, Bi, and Cu, which would affect the accumulation and stripping of Pb(II). On the other hand, the intermetallic compounds of Cu-Bi-Pb [16,44,45] might compete for the active sites on the WE surface and further hinder the electroreduction of Pb^2+^ on the electrode and the electro-oxidation of Pb^0^ off the electrode surface during the SWASV measurement.

The calibration unitary linear models between the stripping current peaks of Pb(II) and Pb(II) concentration in the presence of different Cu(II) concentrations were calculated to obtain the slopes and intercepts, as shown in Table 1. The standard errors, *R*^2^, and *p*-values were analyzed. At a specific Cu(II) concentration, the calibration models exhibited excellent performance—i.e., the values of *R*^2^ were more than 0.94 and the *p*-values were less than 0.01—which suggested that these models had a strong detection capability and remarkable significance. Although unitary linear models with good performance could be achieved, these linear models for Pb(II) detection obviously varied with the change in Cu(II) concentration, which could be determined from different rows of Table 1. In the actual sample detection, the concentration of Cu(II) was unknown; hence, we could not select a suitable unitary linear model to detect the Pb(II) concentration. Therefore, a high-accuracy model for Pb(II) detection in the presence of different Cu(II) contents should be established.

### 3.5. Masking Effect of Ferricyanide on the Cu(II) Interference

Cu(II), as a proverbial interference ion, would hinder the stripping voltammetry signals of Pb(II), especially on a bismuth film-modified electrode. The interference mechanism was elucidated as follows: Cu(II) would be preferentially electroplated on the electrode surface compared with the Bi and Pb, and the above metal ions would form mixed intermetallic compounds [11,12,13,46], which could severely interfere with the detection of Pb(II) by using SWASV method on the bismuth-film modified electrode.

Previous reports have indicated that the effect of Cu(II) on the stripping voltammetry signals of Pb(II) could be reduced by adding ferrocyanide ions because the ferrocyanide ions could react with Cu^2+^ to form a stable complex [14,15,47], which could greatly decrease the number of free Cu^2+^ and/or Cu ions. Therefore, 0.25 μM ferrocyanide was first applied to relieve the effect of 15 μg/L of Cu^2+^ on 15 μg/L of Pb^2+^, showing a weak improvement in the stripping peak current of Pb(II) and a small reduction in the stripping current of Cu^2+^, as shown in Figure 6. To obtain an enhanced result, 1.25 μM ferrocyanide was added to the solution to be tested, demonstrating an obvious enhancement in the stripping current of Pb^2+^ and a large reduction in the stripping current of Cu^2+^.

The above analysis results revealed that ferrocyanide was unideal to eliminate the interference of Cu(II) unless its concentration was optimized for specific samples to be test. The optimization process of ferrocyanide concentration significantly increased the lead detection time because the copper ions were ubiquitous in the soil environment and varied in content.

### 3.6. Establishment of an SVR Model for the Pb^2+^ Detection in the Presence of Cu^2+^

To overcome the disadvantage of using ferrocyanide ions to mask the interference of Cu^2+^ on the SWASV detection for Pb(II), in this study, an SVR model was established with the stripping peak currents of Pb*^2+^* and Cu^2+^ as inputs to detect the concentration of Pb^2+^ in the presence of Cu^2+^ accurately.

#### 3.6.1. Optimization of the SVR Model Parameters by Using the Grid Search Algorithm

The SVR model was studied to characterize the nonlinear relationship between the input variables (stripping peak currents of Pb^2+^ and Cu^2+^) and output (Pb^2+^ concentration). The radial basis function served as the SVR kernel function in the study. The parameters of *c* and *g*, all in the search range from 2^−8^ to 2^8^, were optimized using a grid search algorithm. Internal cross validation was performed using fivefold cross validation, and parameters *g* and *c* were determined using the smallest RMSE of the cross-validation dataset. However, there were many combinations of parameters c and g corresponding with the smallest RMSE value, which constituted a contour line. Commonly, the first parameter combination was selected to calculate the SVR model.

Figure 7 shows the optimization results of parameters g and c. The contour line with the optimal RMSE of 0.0781 was obtained: it was located between the contour line with 0.1 and contour line with 0.05. Under the best parameters g and c, which were 194.0117 and 0.1436, respectively, the Pb^2+^ concentration detection model achieved optimal performance with cross-validation RMSE of 0.0781; however, it was not the actual RMSE value of Pb^2+^ concentration because the data in the validation dataset were normalized to the range of [0, 1] during the cross-validation step. In accordance with the inverse normalization rule, the cross-validation RMSE was 1.7573 μg/L. The location of the optimal parameter combination was marked using a red square, as shown in Figure 7. Subsequently, the values of 194.0117 and 0.1436, serving as the parameters of g and c, respectively, were used to build a nonlinear model for accurate detection of Pb^2+^ concentration.

#### 3.6.2. Analysis of the SVR Model Result for Pb(II) Concentration Detection

The optimized parameter combination was applied to establish the SVR detection model for Pb(II) concentration. A total of 50 experimental data samples as the training dataset were used to train the SVR model, and 20 groups of samples as the test dataset were used to examine the detection capability and stability of the established SVR model. To analyze the detection performance of the developed SVR model, the model detection results, including *R*^2^ and RMSE of the training and test datasets, are presented in Appendix A. The values of *R*^2^ and RMSE of the training dataset were 0.9954 and 0.9891 μg/L, respectively; the values of R^2^ and RMSE of the test dataset were 0.9942 and 1.1204 μg/L, respectively. The *R*^2^ value of the training and test datasets were approximate and all larger than 0.99, demonstrating that the proposed SVR model possessed strong stability and excellent detection capability for Pb(II) concentration.

A multiple linear regression model was established with the stripping peak currents of Pb^2+^ and Cu^2+^ serving as inputs to verify the nonlinear effect of Cu^2+^ on the stripping current of Pb(II) further. The results of the established multiple linear regression model, as shown in Table 2, were as follows: the values of *R*^2^, RMSE, and RSD of the training dataset were 0.4396, 9.9961 μg/L, and 165.7286%, respectively; the values of *R*^2^, RMSE, and RSD of the test dataset were 0.4283, 10.9318 μg/L, and 230.3635%, respectively. These results displayed poor performance for Pb(II) concentration detection. As shown in Table 2, the RMSE values of the multiple linear regression model were approximately 10 times the RMSE values of the SVR model, but the *R*^2^ values of the multiple linear regression model were less than half of the *R*^2^ values of the SVM model on the training and test datasets. This finding indicated that the SVR model had much better detection performance than the multiple linear regression model, which was likely due to the nonlinear effect of Cu^2+^ concentration on the stripping peak current of Pb(II).

The linear relationships between true Pb(II) concentrations and outputs (predicted Pb(II) concentration) of the multiple linear regression and SVR models are presented in Figure 8, and the detection results of Pb(II) concentration under different Cu(II) concentrations are highlighted by scattered points and dotted lines with various colors, which clearly shows the great difference between the multiple linear regression and SVR models for Pb(II) concentration detection. Compared with Figure 8a,b, the scattered points in Figure 8c,d are uniformly and compactly distributed on both sides of the fitting lines, indicating that the predicted values of the SG-SVR model were strongly correlated with the true values compared with those of the multiple linear regression model. Furthermore, the multiple linear model exhibited good detection capacity only at a specific concentration of Cu(II); however, the SVR could accurately detect the Pb(II) concentration in the presence of various concentrations of Cu(II). This result further illustrated that the Pb(II) concentration had a nonlinear relationship with the stripping peak currents of Pb(II) and Cu(II), and this nonlinear relationship could be accurately regressed using the radial basis function kernel function of SVR.

Moreover, the detection performance of the proposed method in this study was compared with the others published in previous papers, as shown in Table 3. This SVR model had the smaller RSD value (7.19%) than other elimination methods of Cu(II) interference. The RSD indicated the average difference between the true and the predicted Pb(II) concentrations. Accordingly, the detection method of coupling SWASV with SVR, which required lower cost and simpler electrode modification, could accurately detect the Pb(II) concentration under the interference of Cu(II).

The interference of other metal cations existing commonly in the actual soil environment, such as Zn^2+^, Hg^2+^, Cd^2+^, Cr^2+^, K^+^, Na^+^, and As^3+^, with the stripping signals of Pb^2+^ was studied at the concentration ratio of 1:1, in which the concentrations of Pb^2+^ and other ions were 15 μg/L. The interference level of the above ions with the stripping peak current of Pb^2+^ was expressed using the relative standard deviation (RSD) value. The results showed that the RSD values of the above ions were all less than 10%, as shown in Appendix A. In detail, the existence of Zn^2+^ increased the stripping peak current of Pb^2+^, and the other metal ions decreased the Pb^2+^ signal.

### 3.7. Validation of the Proposed SVR Model by Using Real Soil Sample Extracts

Real soil sample extracts were prepared in accordance with Section 2.5 to verify the applicability of the proposed SVR coupled with the SWASV method. The stripping peak currents of Pb^2+^ and Cu^2+^, as presented in Table 4, were substituted into the trained SVR model to predict the concentration of Pb(II). The results were compared with the Pb(II) concentration detected using the standard addition method (SAM) and AAS method, as shown in Table 4. Although SAM had an excellent average recovery rate of 99.12%, the detected concentrations of Pb(II) by using SAM were less than those detected using AAS—a recognized method for accurate detection of HMs concentration [50,51]. From Table 4, the stripping peak currents of Cu^2+^ varied in different samples, which demonstrated that Cu^2+^ existed in soil sample extracts and that the Cu^2+^ concentrations varied. Under the interference of Cu^2+^, the stripping peak current of Pb(II) decreased, which in turn caused the low concentrations detected using SAM. For the three soil samples, the RMSE value between the detection results of SAM and AAS was 4.5139 μg/L, but the RMSE value between the detection results of SVR and AAS was only 0.2135 μg/L. Compared with SAM, the SVR method possessed approximately equal detection results to AAS and a satisfactory average recovery result of 98.70%. Moreover, the RMSE value (0.9942 μg/L, as presented in Table 2) of the developed SVR model was far less than the Pb(II) concentration (more than 5.54 μg/L, as presented in the AAS results of Table 4) in real soil samples collected in this study. The above results indicated that SVR could optimize the SWASV detection result of Pb(II) under the disturbance of Cu^2+^ and that the combination of SVR and SWASV could be used to detect Pb(II) in soil samples accurately. Moreover, the electrochemical detection performance of the method proposed in this study, which required shorter detection time, lower cost, and simpler electrode modification, was even on a par with previous works [52,53].

## 4. Conclusions

A new method that coupled a machine learning algorithm with SWASV was developed in this paper to detect Pb(II) concentration quantitatively to overcome the negative effect of Cu^2+^ on the SWASV detection of Pb(II). The key contribution of this work was the accurate detection of Pb(II) concentration under the interference of Cu^2+^ without complex electrode modification and adding a masking reagent (e.g., ferrocyanide). Moreover, this paper not only paid attention to the modeling itself but also concerned the analysis of causative input variables of the model, illustrating the characteristics of SWASV and the interference of different Cu^2+^ concentrations on the stripping peak currents of Pb^2+^. A comparison of the statistical results (*R^2^*, RMSE) of the training and test datasets of the multiple linear regression and SVR models demonstrated that a nonlinear regression could better characterize the potential relationship of the two inputs (i.e., stripping peak currents of Pb^2+^ and Cu^2+^) and output (Pb^2+^ concentration) and showed that the SVR model had better detection accuracy. Furthermore, real soil samples were applied to verify the detection performance of the developed SVR model by comparing with the results of the AAS method. Compared with SAM, SWASV-SVR (the RMSE with AAS was 0.2135 μg/L) had an approximately equal detection concentration to AAS and a satisfactory average recovery rate of 98.70%, confirming the practicability of the developed method. The developed model was expected to be embedded in an electrochemical analysis device for accurately detecting Pb(II) in the presence of Cu^2+^. This method, coupling SWASV with a machine learning regression algorithm, can provide a new idea for accurate detection of the concentration of HMs for environmental monitoring or food safety supervision.

In real soil sample detection, a dataset with the wide concentration range of Pb(II) and Cu(II) needs be prepared to ensure that the concentration of Pb(II) and Cu(II) in soil was within the range of the dataset. Then, the stripping peak currents of Pb(II) and Cu(II) could be instituted into the established to predict the concentration of Pb(II). Furthermore, more soil samples need be collected to prove the performance of the proposed method.

## Figures and Tables

**Figure 1 sensors-20-06792-f001:**
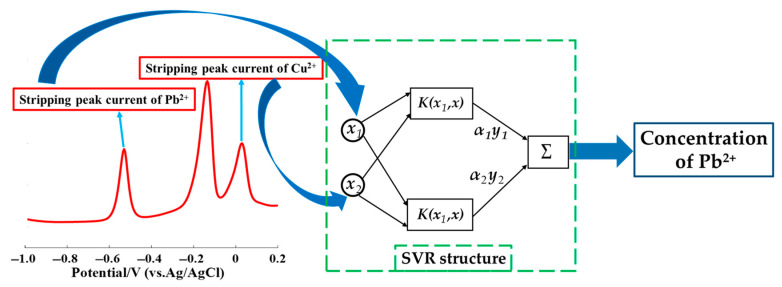
Schematic of the support vector regression (SVR) model used to detect the Pb^2+^ concentration in the presence of Cu^2+^.

**Figure 2 sensors-20-06792-f002:**
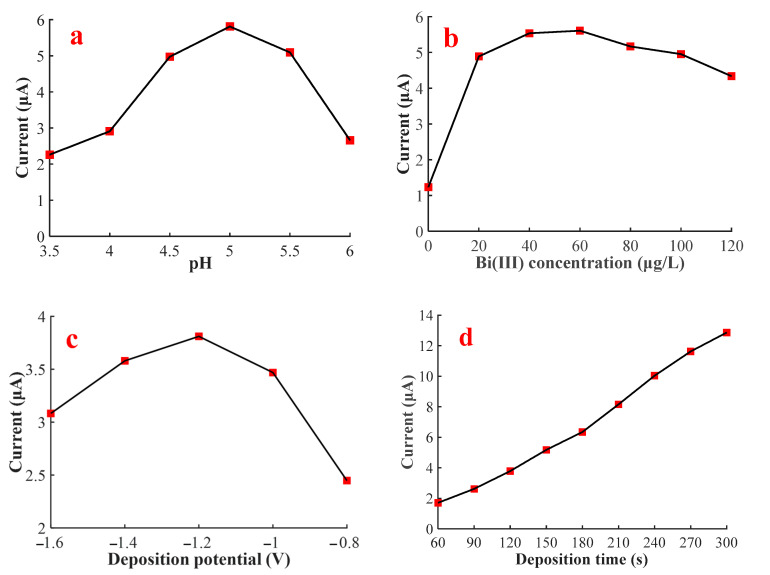
Effects of pH value, Bi(III) concentration, deposition potential, and deposition time on the stripping peak current of 10 μg/L of Pb(II). (**a**) Optimization of pH value in the presence of 100 μg/L of Bi(III) ions; (**b**) optimization of Bi(III) concentration at a pH value of 5.0; (**c**) optimization of deposition potential at a pH value of 5.0, Bi(III) concentration of 60 μg/L, and deposition time of 120 s; (**d**) optimization of deposition time at a pH value of 5.0, Bi(III) concentration of 60 μg/L, and a deposition potential of −1.2 V.

**Figure 3 sensors-20-06792-f003:**
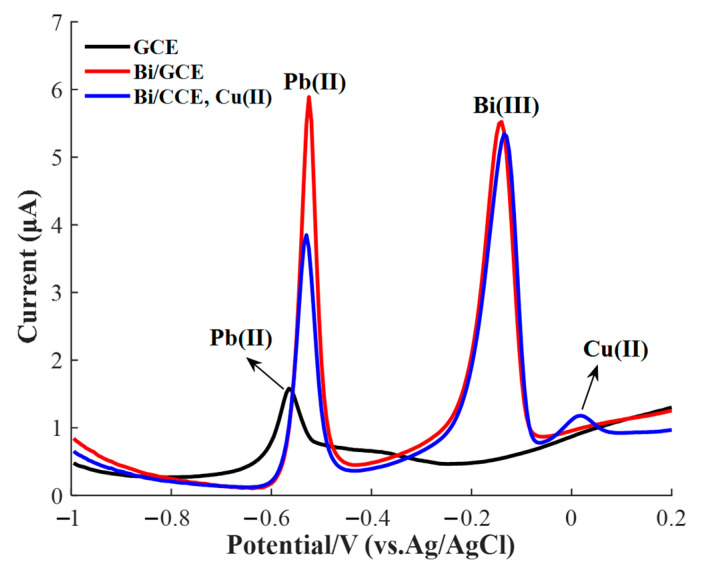
Square wave anodic stripping voltammetry (SWASV) signals of 10 μg/L of Pb(II) in 0.2 M acetate acid buffer solution on bare GCE, Bi/GCE, and Bi/GCE in the presence of 5 μg/L of Cu(II). pH value of 5.0; Bi(III) concentration of 60 μg/L; deposition potential of −1.2 V; deposition time of 150 s.

**Figure 4 sensors-20-06792-f004:**
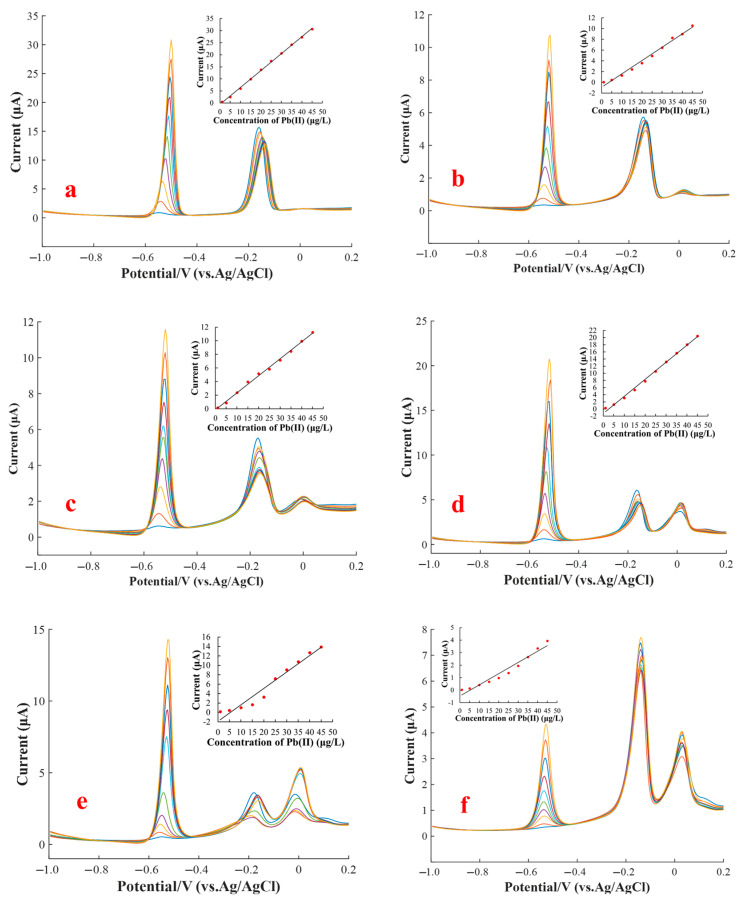
SWASV measurement signals for Pb(II) ranging from 1.0 to 45 μg/L in different Cu(II) concentrations: (**a**) 0, (**b**) 1, (**c**) 5, (**d**) 10, (**e**) 15, (**f**) 20, (**g**) and 25 μg/L. pH value of buffer solution: 5.0; Bi^3+^ concentration: 60 μg/L; deposition potential: −1.2 V; deposition time: 150 s.

**Figure 5 sensors-20-06792-f005:**
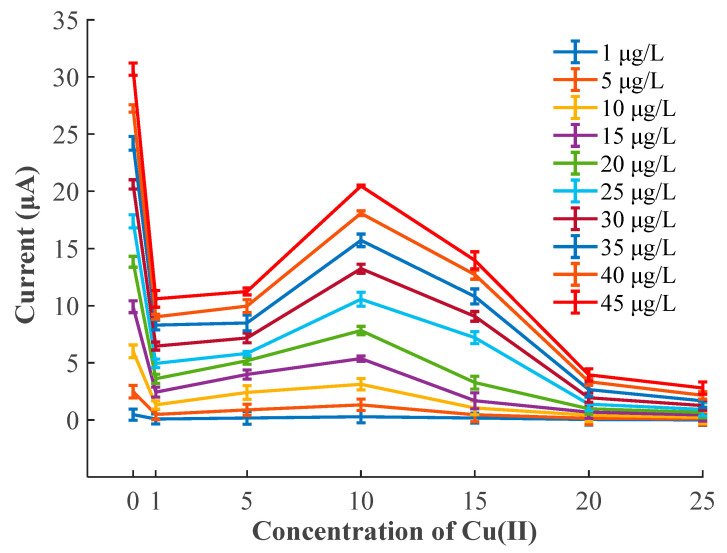
Interference of various Cu(II) concentrations on the stripping current peaks of Pb(II).

**Figure 6 sensors-20-06792-f006:**
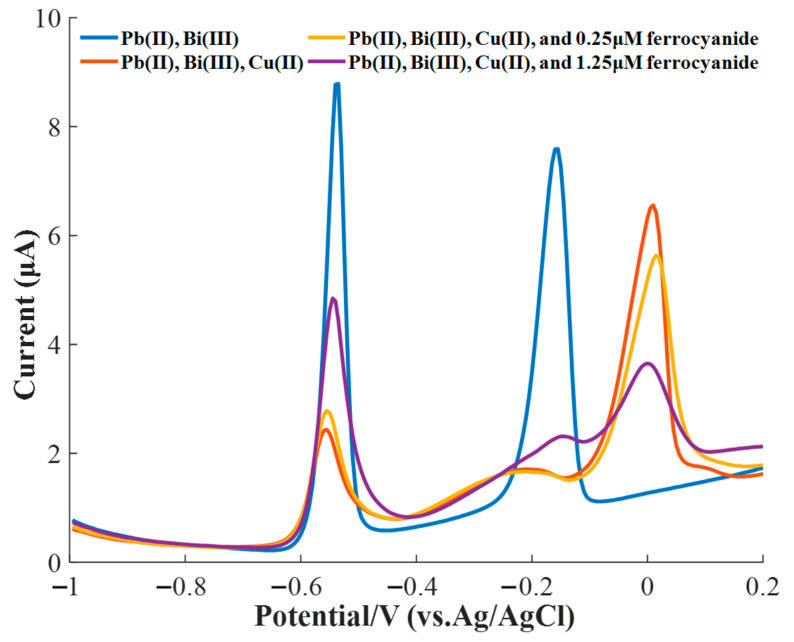
Masking effect of different concentrations of ferricyanide on the interference of 15 μg/L of Cu(II) for the SWASV measurement of 15 μg/L of Pb(II).

**Figure 7 sensors-20-06792-f007:**
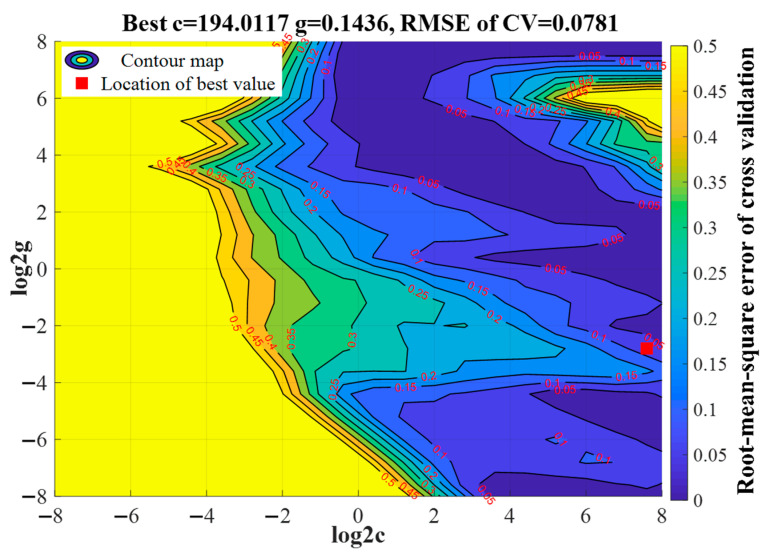
Optimization result of c and g parameters by using the grid search algorithm.

**Figure 8 sensors-20-06792-f008:**
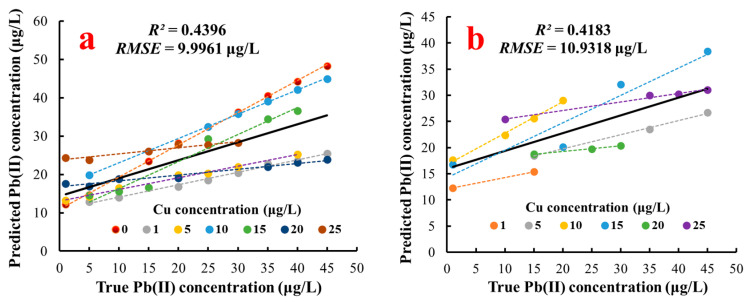
Prediction results of Pb(II) concentration. (**a**) Training set of multiple linear regression; (**b**) test set of multiple linear regression; (**c**) training set of SVR; (**d**) test set of SVR.

**Table 1 sensors-20-06792-t001:** Calibration unitary linear models of Pb(II) stripping peak current and concentration in different concentrations of Cu(II).

Concentration of Cu(II) (μg/L)	Calibration Unitary Linear Model of Pb(II)				*R* ^2^	*p*-Value
	Slope		Intercept			
	Coefficients	Standard Error	Coefficients	Standard Error		
0	0.7011	0.0160	−0.5645	0.2974	0.9989	9.56 × 10^−6^
1	0.2484	0.0095	−0.9076	0.2542	0.9884	5.01 × 10^−6^
5	0.2497	0.0054	−0.1332	0.1435	0.9963	5.10 × 10^−6^
10	0.4766	0.0111	−1.1908	0.2961	0.9957	9.5 × 10^−7^
15	0.3496	0.0259	−1.8937	0.6900	0.9581	8.57 × 10^−5^
20	0.0896	0.0069	−0.4743	0.1857	0.9540	1.25 × 10^−5^
25	0.0608	0.005	−0.3550	0.1427	0.9417	3.23 × 10^−5^

**Table 2 sensors-20-06792-t002:** Statistics of the Pb(II) detection results of the multiple linear regression and SVR models.

Model	Training Dataset			Test Dataset		
	*R* ^2^	RMSE (μg/L)	RSD	*R* ^2^	RMSE (μg/L)	RSD
Multiple linear regression	0.4396	9.9961	165.7286%	0.4183	10.9318	230.3635%
SVR	0.9954	0.9891	7.1973%	0.9942	1.1204	7.4282%

**Table 3 sensors-20-06792-t003:** The interferences and eliminating methods of Cu(Ⅱ) on Pb(Ⅱ) detection using different electrodes.

Electrode	RSD	Eliminating Method	Reference
GC/GQDs-NF/GCE ^1^	35.10%	Adding 0.1 mM ferrocyanide	[12]
(BiO)_2_CO_3_@SWCNT-Nafion/GCE ^2^	20.54%	Adding 0.1 mM ferrocyanide	[13]
SWCNTs-Nafion/IL/SPE ^3^	27.26%	Adding 0.1 mM ferrocyanide	[22]
Bi/SPE ^4^	45.00%	Optimizing the added concentration of ferricyanide	[45]
Bi/p-Tyr/GCE ^5^	49.91%	Electrodeposition of Cu^2+^ for 35 min prior to detection of Pb^2+^	[48]
Bi_2_O_3_/GCE ^6^	20.00%	Adding 0.1 mM ferrocyanide	[49]
Bi-film/GCE ^7^	7.19%	Combining with support vector regression	This work

Footnote: ^1^ GC/GQDs-NF/GCE: Graphene quantum dots-nafion modified glassy carbon electrode; ^2^ (BiO)_2_CO_3_@SWCNT-Nafion/GCE: (BiO)_2_CO_3_@single-walled carbon nanotube nanocomposite/nafion composition modified glassy carbon electrode; ^3^: SWCNTs-Nafion/IL/SPE: Bi/single-walled carbon nanotubes-nafion/ionic liquid nanocomposite modified screen-printed electrode; ^4^ Bi/SPE: Bismuth film modified screen-printed carbon electrode; ^5^ Bi/p-Tyr/GC: rod-like poly-tyrosine/Bi modified glassy carbon electrode; ^6^ Bi_2_O_3_/GCE: Bismuth oxide modified glassy carbon electrode; ^7^ Bi-film/GCE: Bismuth film modified glassy carbon electrode.

**Table 4 sensors-20-06792-t004:** Results of the detection of Pb(II) concentration in soil sample extracts by using standard addition method (SAM), the proposed SVR model, and atomic absorption spectroscopy (AAS).

Sample Number	Added (μg/L)	Stripping Peak Current (μA)		Detected Concentration (μg/L)			Recovery (%)		
		Pb^2+^	Cu^2+^	SAM	SVR	AAS	SAM	SVR	AAS
1	-	0.88	0.55	2.81	5.42	5.54	-	-	-
	5	2.40	0.72	7.75	10.66	10.78	98.80	104.80	104.80
	10	3.99	0.77	12.84	14.54	14.86	100.30	91.20	93.20
2	-	5.37	2.82	10.23	15.55	15.85	-	-	-
	10	10.56	3.06	20.17	24.78	25.72	99.40	92.30	98.70
	15	13.23	3.13	24.98	30.52	30.97	98.33	99.80	100.80
3	-	0.42	2.47	5.25	9.77	9.95	-	-	-
	15	1.38	2.39	20.07	25.04	25.18	98.80	101.80	101.53
	20	1.95	2.40	25.07	30.23	30.31	99.10	102.30	101.80

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
