# Peer review of "Coupling Square Wave Anodic Stripping Voltammetry with Support Vector Regression to Detect the Concentration of Lead in Soil under the Interference of Copper Accurately"

_sensors, 2020, doi:10.3390/s20236792_

Round 1

Reviewer 1 Report

Manuscript sensors-1014190 by Ning Liu et al. reports on an approach to mathematically overcome the interferences of Cu(II) in electrochemical detection of Pb(II) ions. Even though the experimental work itself does not present a breakthrough, the novel evaluation method is worth publishing and will find its readers. However, before the manuscript can be recommended for acceptance, major revisions are necessary, as indicated below.

1. The discussion is practically missing in the manuscript. The authors should compare the achieved results with other literature reports.

2. It should be proved experimentally whether the presence of Cu(II) affects the accumulation or the stripping step. Pb(II) should be first accumulated without the presence of Cu(II), followed by the addition of Cu(II) and stripping analysis to explain the mechanism in-depth.

3. The limit of detection of the method should be determined and compared to the typical concentrations of Pb(II) in soil samples.

4. Closer looks at Figure 10 reveals a pattern that might correspond to two independent linear relationships instead of a single one with a larger error. The authors should try plotting the Cu(II) concentration as the varying color of the points; this might reveal some trend.

5. The discussion of interference of other metal ions should be accompanied by data (in Supporting Information).

6. It should be clearly described how the optimal conditions in Figure 8 were chosen. According to the figure, there is a broad zone that provided a low RMS error.

7. Figure 2 does not provide any valuable information and should be either deleted or moved to Supporting Information.

8. What is the advantage of Figure 9 over the representation of data in Figure 10? The authors should either clearly highlight the importance of Figure 9 or remove it. Also, connecting the points in Figure 9 does not seem correct.

9. The experimental conditions of the AAS measurement should be provided in the Materials and Methods section.

10. The manuscript contains many abbreviations, which are not familiar to readers of an analytical journal (e.g., RFB, GS, MLR). Even though they are correctly explained on their first occurrence, I would consider writing the full names (e.g., “grid search” is not that long), facilitating the reading for analytical chemists.

11. Even though the level of English is acceptable, I would highly recommend proofreading by a native speaker to improve the readability.

12. Several minor edits are necessary.
i. “hydrogen protons” (page 5, line 216) should be replaced by “protons”.
ii. Is the description of peaks as “weak and small” and “strong and large” necessary? Merely writing “small” and “large” will be sufficient.
iii. The quotation marks for “red square” (page 11, line 358) are not necessary.
iv. “cation ions” (page 13, line 405) should be replaced by “cations”.
v. References should be corrected (e.g., author name in ref. 25; “ence” in ref. 30; journal name in capital letters in refs. 31 and 35; “nafifion” in ref. 42).

Reviewer 2 Report

See attached file

Round 2

Reviewer 1 Report

The authors have successfully addressed my comments. The manuscript can be accepted in the present form.

Reviewer 2 Report

All the comments and suggestions that I mentioned have been responded and revised. I recommend for publishing.